# A Tiny Object Detection Approach for Maize Cleaning Operations

**DOI:** 10.3390/foods12152885

**Published:** 2023-07-29

**Authors:** Haoze Yu, Zhuangzi Li, Wei Li, Wenbo Guo, Dong Li, Lijun Wang, Min Wu, Yong Wang

**Affiliations:** 1Beijing Advanced Innovation Center for Food Nutrition and Human Health, College of Engineering, China Agricultural University, 17 Qinghua Donglu, P.O. Box 50, Beijing 100083, China; yuhaoze141800@126.com (H.Y.); 17826075723@163.com (W.L.); gwb0725@126.com (W.G.); minwu@cau.edu.cn (M.W.); 2School of Electronic and Computer Engineering, Peking University, Shenzhen 518055, China; lizhuangzi@stu.pku.edu.cn; 3Beijing Key Laboratory of Functional Food from Plant Resources, College of Food Science and Nutritional Engineering, China Agricultural University, Beijing 100083, China; 4School of Chemical Engineering, University of New South Wales, Sydney, NSW 2052, Australia; benjaminwy@gmail.com

**Keywords:** cleaning operation, maize image, tiny object detection, feature integration

## Abstract

Real-time and accurate awareness of the grain situation proves beneficial for making targeted and dynamic adjustments to cleaning parameters and strategies, leading to efficient and effective removal of impurities with minimal losses. In this study, harvested maize was employed as the raw material, and a specialized object detection network focused on impurity-containing maize images was developed to determine the types and distribution of impurities during the cleaning operations. On the basis of the classic contribution Faster Region Convolutional Neural Network, EfficientNetB7 was introduced as the backbone of the feature learning network and a cross-stage feature integration mechanism was embedded to obtain the global features that contained multi-scale mappings. The spatial information and semantic descriptions of feature matrices from different hierarchies could be fused through continuous convolution and upsampling operations. At the same time, taking into account the geometric properties of the objects to be detected and combining the images’ resolution, the adaptive region proposal network (ARPN) was designed and utilized to generate candidate boxes with appropriate sizes for the detectors, which was beneficial to the capture and localization of tiny objects. The effectiveness of the proposed tiny object detection model and each improved component were validated through ablation experiments on the constructed RGB impurity-containing image datasets.

## 1. Introduction

The performance of the cleaning system is of paramount importance, as it is a critical step in combined harvesting. It exerts a direct influence on the loss rate and impurity content of grain kernels, while also playing a vital role in ensuring efficient drying, quality-guaranteed transportation and safe storage of the harvested grains [1,2]. The cleaning principles are often based on the significant differences in shape, specific gravity, volume and density, etc. among normal kernels, damaged ones, rotten ones and impurities. The principle involves the actions of throwing, blowing, transporting and screening the mixture, through the coupling of multiphysics [3,4]. For this purpose, Krzysiak et al. proposed a rotary cleaning device suitable for wheat grains and analyzed the influences of the inclination angle of the sieve drum on the quality of the processes. Three metrics were used to assess the cleaning results: the coefficients of plump grain mass separation and fine impurity separation and the overall coefficient of cleaning effectiveness [5]. On the premise of sufficiently considering the air-flow uniformity in the technological processes of grain air-screen cleaning, Aldoshin et al. installed an additional fine-mesh sieve between the lower sieve and inclined bottom of the cleaning system to isolate the small impurities [6]. The countersunk screen designed by Wang et al. was utilized in the cleaning device so that the maize particles could move towards the screen holes, which increased the penetration possibility of maize kernels [7]. These contributions optimized the mechanical structure of the equipment based on the physical characteristics of different varieties of crops, which could improve the cleaning efficiency to a certain extent. However, the parameter setting process still relied on operators to manually track and supervise the entire cleaning processes, relying on their experience. This approach was evidently time-consuming and expensive. Therefore, the implementation of real-time grain situation awareness can offer valuable guidance and reference for the adaptive and dynamic adjustments of cleaning strategies, addressing these limitations. 

As a kind of information carrier, images can provide research foundation and data resources for numerous fields [8]. Based on the hyperspectral, a rapid and cost-effective way was proposed to generate records of sediment properties and composition at the micrometer-scale [9]. Yuan et al. designed a compact proxy-based deep learning framework to perform highly accurate hyperspectral image classification with superb efficiency and robustness [10]. In addition, the maize kernel images supplied information support to the classification tasks of planted cultivars [11]. Object detection, through the integration of object localization and recognition techniques, enables accurate regression of bounding box coordinates and identification of object categories. This approach was widely applied in the domains of face recognition, medical image processing and agricultural product processing, etc. [12,13]. The traditional object detection algorithms devised the corresponding feature extraction modules for different kinds of objects to be detected, so they were more pertinent and interpretable [14]. Nevertheless, these methods exhibited limitations in terms of robustness and scalability. This was primarily attributed to their heavy dependence on manually crafted features and the need for extensive parameter adjustments [15,16]. Relying on the powerful feature extraction capabilities, deep learning-based object detection technologies could adaptively capture the deep semantic information of images through the multi-structured network models, thus significantly improving the efficiency and accuracy of detection tasks [17,18]. Wang et al. constructed Pest24, which was a large-scale multi-target standardized dataset of agricultural pests. On this basis, they utilized a variety of deep learning-based object detection models to detect the pests in the datasets, which achieved encouraging results in the real-time monitoring of field crop pests [19]. Based on the deep neural network frameworks, Bazame et al. proposed a computer vision system with the object detection algorithms as the core to measure the ripeness of Coffea Arabica fruits on the branches, thereby demonstrating the potential in objectively guiding the decision-making of the coffee farmers [20]. As one of the classic two-stage detectors, Faster Region Convolutional Neural Network (Faster R-CNN) could be used to identify the weeds in cropping areas and detect the cracks in bridges [21,22]. Compared with the representative contributions of the one-stage algorithms You Only Look Once (YOLO) [23,24] and Single-Shot Multi-Box Detector (SSD) [25,26], due to the initial generation of the candidate box and the further adjustment of the bounding box, the detection accuracy of the two-stage models was relatively higher, while the one-stage models had faster detection speed. In order to comprehensively detect all kinds of objects with different geometric characteristics in the images, the multi-feature fusion based on convolution, the setting of residual module and the introduction of attention mechanism were exerted to the basic framework of the backbones, which gradually complicated the structure of the feature learning networks. A feature pyramid architecture AugFPN was designed by Guo et al. to realize the fusion of multi-scale image features; the ResNet50 and MobileNet-v2 were employed as the backbone respectively to demonstrate its effectiveness on the MS COCO detection datasets [27]. For the purpose of capturing the rich context features of the image to be detected, Zhao et al. proposed a context-aware pyramid feature extraction module (CPFE) for the high-level feature maps. At the same time, the enhancement of contextual features and the refinement of boundary information (contained in the low-level feature maps) were realized with the aids of the channel-wise attention and spatial attention, and the final matrix was generated through feature fusion [28]. 

Limited by the lack of visual feature information caused by fewer pixels, the detection accuracy of tiny objects was relatively low [29,30]. In addition, the information loss during the forward propagation of the networks, the uneven distribution of the sample quantities and the setting of anchor boxes, etc. could all affect the final object classification and the coordinate regression results [31]. Therefore, tiny object detection has become one of the most challenging tasks in computer vision [32]. In allusion to the smaller size and higher density of the objects in the aerial images, Wei et al. proposed an efficacious calibrated-guidance (CG) scheme to intensify the channel communication through the feature transformer fashion, which could adaptively determine the calibration weights for each channel based on the global feature affinity correlations [33]. The concept of fusion factor was proposed by Gong et al. to control the information that delivered from deep layers to the shallow ones, which adapted the feather pyramid network (FPN) to tiny object detection, and its effective value was estimated based on a statistical method [34]. By use of the improved K-means clustering algorithm, Wang et al. generated suitable anchors for the traffic sign datasets and then promoted the detection recall rate and target positioning accuracy of the proposed lightweight recognition algorithm, which was improved on the basis of YOLOv4–Tiny [35]. Similarly, Cheng et al. adjusted the sizes and aspect ratios of the anchors and label frames according to the dimensioning of the tiny objects in the capacitance samples, thereupon achieving effective training of the network in the candidate areas [36]. In addition, different data augmentation strategies had been testified to expand and enrich the scale and diversity of the datasets, thus enhancing the robustness and generalization ability of the detection models [37]. 

Maize (*Zea mays* L.) is a traditional global grain crop known for its strong environmental adaptability, high nutritional value and diverse applications. It serves as a crucial feed source in the animal husbandry and breeding industry [38,39]. As a consequence, the rational utilization of maize production capacity had momentous strategic significance for the development of national economy and the promotion of agricultural technology [40]. However, during the harvesting process, maize kernels often become contaminated with a variety of impurities, including rotten and damaged kernels, cobs, husks, gravel and clods. These result in resource waste and pose safety hazards during subsequent processing and storage [41]. Hence, this study focused on harvested maize as the primary material and introduced a tiny object detection network specifically designed for impurity-containing maize images. This network enabled real-time identification and analysis of impurity categories and their distribution during cyclic cleaning operations. By utilizing the feedback on grain conditions during impurity removal, targeted and dynamic adjustments of parameters and strategies could be made to enhance the efficiency and minimize losses in the maize cleaning process. The major contribution points are summarized as follows:(1)The EfficientNetB7 was introduced as the backbone of the feature learning network and a tiny object detection network was proposed for analyzing the categories and distribution of impurities in the harvested maize based on the classic contribution Faster R-CNN;(2)The designed cross-stage feature integration mechanism was able to fuse the semantic descriptions and spatial information of feature matrices from different hierarchies through continuous convolution and upsampling operations;(3)Based on the geometric properties of the objects to be detected and the resolution of images, the adaptive region proposal network was able to generate appropriate anchor boxes for the detectors;(4)The impurity-containing maize datasets was constructed to measure the comprehensive performance of the end-to-end tiny object detection network.

## 2. Materials and Methods

The variety of maize in this research was Wannuo 2000, which was purchased from Shangzhuang experimental station of China Agricultural University (Beijing, China). The moisture content was about 25% and the samples were stored in the refrigerator at 4 °C. Figure 1 revealed the overall framework of tiny object detection for the impurity-containing maize images and it could be divided into three parts, which were image feature learning network, adaptive region proposal network and classification and regression layers of candidate box, according to the propagation sequence. The image feature learning network was used to extract the global features that contained multi-scale mappings. The adaptive region proposal network performed coordinate adjustment and classification of generated anchor boxes through continuous convolution. Eventually, the obtained high-quality candidate boxes were subjected to specific classification and location regression.

### 2.1. Image Feature Learning Network 

EfficientNet has marked a significant milestone in compound model scaling research by effectively balancing network width, depth and resolution. This balance enables the models to sufficiently capture the feature of images, while simultaneously making them more effortless to be trained [42]. Therefore, based on the fine-grained object detection task, EfficientNetB7 was introduced as the backbone of the image feature learning network. In the feed-forward processes of the model, compared with the feature matrices from the deep hierarchies, those from shallow hierarchies contain abundant spatial information but exhibit relatively ambiguous semantic descriptions [43]. Therefore, the cross-stage integration mechanism shown in Figure 2 was embedded in the basic framework of EfficientNetB7. By performing convolution and upsampling operations on the feature matrices from deep hierarchies and fusing them with those from shallow hierarchies, a cross-stage integrated feature with multi-scale mappings was acquired [44]. Among them, the convolution operations with different receptive fields could simultaneously improve the expression ability of the model and adjust the dimension of the feature matrices.

The feature learning of the impurity-containing maize images was conducted through eight convolution stages; as shown in Table 1, the width and depth of each stage were closely related to the dimension of the original images, which were obtained by multiplying the magnification factor corresponding to the resolution with the parameters of the baseline (EfficientNetB0) [45,46] (where $H_{i}*W_{i}*C_{i}$ are the dimensions of the feature matrix before operation $O_{i}$ in Figure 2). $L_{i}$ denotes the quantity of repetitions of the operation $O_{i}$, i.e., the depth of stage i. The rightmost column lists the kernel sliding strides of the first convolutions in the repeated operations for each stage. Compared with the subsequent stages, the operations in the first stage adapted a traditional convolution with a kernel size of 3*3. Furthermore, the incorporation of BN (Batch Normalization) layers and Swish activation functions effectively addressed gradient vanishing and exploding issues during back-propagation, thereby enhancing the model’s generalization capability [47,48].

The detailed structure of MBConv in Stages 2–8 is exhibited in Figure 3, which shows the close layouts with the MobileNetV3 blocks [49]. The first convolution operation, with a kernel size of 1*1, was utilized to increase the dimension of the input feature matrix. MBConv6 in Table 1 signified that the scale of convolution kernels was 6 times that of the input feature channels, while MBConv1 indicated that there was no 1*1 convolution operation of dimensionality enhancement in the current stage. Similarly, k3*3 and k5*5 were the convolution kernel sizes for the depthwise convolution in the corresponding stage [50]. The utilization of depthwise convolution effectively reduced the quantity of network parameters, which meant less memory consumption and faster computing speed. The padding of the 3*3 and 5*5 kernels were 1 and 2 respectively, which meant that the matrix size and channel quantity of the feature did not change after the planar convolution with a stride of 1. Furthermore, it was a necessary and sufficient condition for the existence of shortcut connections and dropout layers that the input and output feature matrices in Figure 3 had the same dimensionality.

The SE block, depicted in Figure 4, serves as a lightweight plug-and-play channel attention mechanism. It compresses features in the spatial dimension using squeeze, excitation and reweight processes. Consequently, based on the correlation among channels, new weights were generated for them and exerted to the input matrices in turn [51]. By virtue of its cross-channel interaction capability, the SE block was able to selectively enhance the more significant features through learning global information [52]. In this case, global average pooling was applied to each channel of the input matrices and Swish and Sigmoid activation functions were utilized for the two one-dimension fully connected layers. The global average pooling downsampled the matrices to the specified size and the activation functions were able to improve the nonlinearity of the network. Different from the SE block in the image classification tasks, the quantity of neurons in the channel-reduced FC Layer 1 was a quarter of the feature width (the quantity of channel) input to the current MBConv. The scale of FC Layer 2 was the same as the feature width after depthwise convolution. With regard to the cross-stage integration mechanism, the convolution operation with a kernel size of 3*3 was exploited to improve the local perception competence of the model and the quantity of 1*1 convolution kernel could flexibly adjust the stacking of channels. Moreover, the double upsampling processes after feature integration were implemented through bilinear interpolation [53].

### 2.2. Adaptive Region Proposal Network (ARPN)

ARPN (Adaptive Region Proposal Network) leverages the distribution characteristics and geometric properties of impurities and maize kernels to classify and adjust the coordinates of generated anchors through continuous convolution. Specifically, the convolution kernel and sliding window with the size of 3*3 were employed to sequentially traverse each position of the cross-stage integrated feature, thereby obtaining the intermediate layer (in the same size and dimension as the cross-stage integrated feature) and generating initial anchor boxes in the meantime [54]. In order to more completely and accurately cover the various objects in the impurity-containing maize images, the aspect ratios were set to 1:1, 1:2 and 2:1, as shown in Figure 5, and the area scales were 64^2^, 128^2^ and 256^2^, which could correspondingly generate about 50K (75*75*9) anchor boxes on each original image [55]. These were determined by conducting experiments on different categories of target contours in the impurity-containing maize images. Eventually, the classification and coordinate regression parameters of each anchor box were attained by concatenating two convolution operations with a kernel size of 1*1. The classification information included the probability of foreground (with object) and background (without object), and the regression parameters were oriented towards the center coordinates, width and height of the anchor boxes, so the quantities of convolution kernels were 2n and 4n, respectively.

In the end-to-end training processes based on the back propagation and stochastic gradient descent, the positive anchor samples were defined as (i) anchors that had IoU (intersection-over-union) overlaps higher than 0.7 with any ground-truth box, or (ii) anchors with the highest IoU ratio with the ground-truth boxes. In contrast, the anchor was regarded as a negative sample when the IoU ratios were lower than 0.3 for all ground-truth boxes [56]. Anchors that were neither positive nor negative did not participate in the updates of the networks. In order to avoid the degradation and poor generalization of the model caused by excessive negative samples, the loss of mini-batch was counted by randomly sampling the equal quantity of positive and negative samples [57]. The loss function of ARPN is shown in Equation (1), which was measured through division of the sum of classification loss and regression loss by the quantity of mini-batch. Among them, $N_{m}=256$ was the capacity of each mini-batch. If the quantity of positive samples was fewer than 128, then the mini-batch was supplemented with negative samples. i represents the index of an anchor in the current mini-batch, $c_{i}$ denotes the probability that the ith anchor was predicted to be the real label. The ground-truth of $c_{i}^{*}$ is 1 if the current anchor box is a positive sample and 0 for a negative sample [58]. $r_{i}^{*}=\{r_{x}^{*},r_{y}^{*},r_{w}^{*},r_{h}^{*}\}$ indicates the coordinate regression parameters of the ith anchor corresponding to the ground-truth box and $r_{i}=\{r_{x},r_{y},r_{w},r_{h}\}$ is the predicted value.
(1)$$Loss(\{c_{i}\},\{r_{i}\})=\frac{\sum_{i}L_{cls}(c_{i},c_{i}^{*})+\sum_{i}c_{i}^{*}L_{reg}(r_{i},r_{i}^{*})}{N_{m}}$$
(2)$$L_{cls}=-\ln(c_{i})$$
(3)$$L_{reg}(r_{i},r_{i}^{*})=\sum_{i}smooth_{L_{1}}(r_{i}-r_{i}^{*})$$

The classification loss $L_{cls}$ and regression loss $L_{reg}$ were separately defined by the logarithmic and cumulative operations of Equations (2) and (3), and the $smooth_{L_{1}}$ revealed in Equation (4), was introduced as a robust loss function [31]. Furthermore, Equation (5) describes the relationships among $Attr_{bef}=\{x_{bef},y_{bef},w_{bef},h_{bef}\}$, $r_{i}$ and $r_{i}^{*}$. $Attr_{bef}$ and $Attr_{aft}=\{x_{aft},y_{aft},w_{aft},h_{aft}\}$ are the attribute information of the anchor and the coordinate-adjusted candidate box, respectively. $Attr_{gt}=\{x^{*},y^{*},w^{*},h^{*}\}$ is the attribute information of the ground-truth box corresponding to the current anchor [59]. The attribute information included the centre coordinates, width and height. The network parameters were randomly initialized through drawing weights from the zero-mean Gaussian distribution with standard deviation of 0.01. Meanwhile, since the cross-boundary anchor boxes brought about a large number of error terms that were difficult to correct, the anchor boxes with boundary-crossing outliers were ignored in the training processes. Finally, based on the classification information of the generated proposal regions, a non-maximum suppression (NMS) approach was adapted to deal with the highly overlapping candidate boxes; the IoU threshold for NMS was fixed at 0.7 [60].
(4)$$smooth_{L_{1}}(x)=\left\{\begin{array}{c} 0.5x^{2}\;\;\;\;\;if\left|x\right|<1\\ \left|x\right|-0.5\;\;\;otherwise \end{array}\right.$$
(5)$$\begin{array}{c} r_{x}=\frac{x_{aft}-x_{bef}}{w_{bef}},\;r_{y}=\frac{y_{aft}-y_{bef}}{h_{bef}},\\ r_{w}=\ln(\frac{w_{aft}}{w_{bef}}),\;r_{h}=\ln(\frac{h_{aft}}{h_{bef}}),\\ r_{x}^{*}=\frac{x^{*}-x_{bef}}{w_{bef}},\;r_{y}^{*}=\frac{y^{*}-y_{bef}}{h_{bef}},\\ r_{w}^{*}=\ln(\frac{w^{*}}{w_{bef}}),\;r_{h}^{*}=\ln(\frac{h^{*}}{h_{bef}}) \end{array}$$

### 2.3. Classification and Regression Layers of Candidate Box

The candidate boxes generated by ARPN served as the regions of interest (ROI) for the follow-up specific classification and location regression. These regions were projected to the cross-stage integrated matrix obtained through the feature learning network [61]. After ROI pooling, the feature matrices were regularized to a consistent size and flattened. Both of the two following fully connection layers had 1024 neurons and were exploited as the inputs of the classifier and regressor. The outputs of the classification layer with *softmax* included k+1 outcomes, which, respectively, represented the probability of objects in different varieties. Among them, k was the quantity of object categories and the circumstances of background were also taken into consideration [62]. Similar to the regression layer in the ARPN, the candidate box regressor contained 4∗(k+1) neurons, which could adjust each location through 4 parameters. As shown in Equation (6), $P=\{P_{x},P_{y},P_{w},P_{h}\}$ represents the center coordinates, width and height of the candidate box, $\left\{U_{x},U_{y},U_{w},U_{h}\right\}$ are the attribute information of the final bounding box output by the tiny object detection network and $\left\{f_{x},f_{y},f_{w},f_{h}\right\}$ are the coordinate regression parameters of k+1 object categories exported by the regressor.
(6)$$\begin{array}{c} U_{x}=P_{w}*f_{x}(P)+P_{x}\\ U_{y}=P_{h}*f_{y}(P)+P_{y}\\ U_{w}=P_{w}*e^{f_{w}(P)}\\ U_{h}=P_{h}*e^{f_{h}(P)} \end{array}$$

The loss of each candidate box in the tiny object detection network was composed of category loss $L_{cat}$ and regression loss $L_{loc}$, as shown in Equation (7) [63]. $q=\{q_{0},q_{1},...,q_{k}\}$ is the *softmax* probability distribution predicted by the classifier, v denotes the real category label corresponding to the object in the candidate box and the category loss $L_{cat}$ is measured through Equation (8). $b^{g}=\{b_{x}^{g},b_{y}^{g},b_{w}^{g},b_{h}^{g}\}$ is the coordinate regression parameters predicted by the regressor for the corresponding category g and $s=\{s_{x},s_{y},s_{w},s_{h}\}$ is that of the candidate box for the corresponding ground-truth object. The regression loss $L_{loc}$ is obtained through Equation (9) and α is the hyper-parameter utilized to balance the two losses [64]. Additionally, the values of the Iverson bracket indicator function [v>0] are 1 when v>0; otherwise, it is 0. Compared with the basic Faster R-CNN network, in order to capture the multi-hierarchy features in the fine-grained impurity-containing maize images, the proposed model replaced the original backbone ZFNet with EfficientNetB7 and embedded a cross-stage feature integration mechanism. At the same time, the area scale of the anchor box in the adaptive region proposal network was also adjusted accordingly for the tiny object detection tasks.
(7)$$Loss(q,v,b^{g},s)=L_{cat}(q,v)+\alpha [v>0]L_{loc}(b^{g},s)$$
(8)$$L_{cat}(q,v)=-\ln q_{v}$$
(9)$$L_{loc}(b^{g},s)=\sum_{i\in \{x,y,w,h\}}smooth_{L_{1}}(b_{i}^{g}-s_{i})$$

## 3. Results and Discussion

The image acquisition modules, as illustrated in Figure 6C, were positioned at the feed port and discharge port of the cleaning equipment. Their purpose was to capture images and provide the necessary data for the end-to-end tiny object detection network. The multiphysics-coupled cleaning equipment removed impurities with a certain mass through two screens with different sizes and shapes, while relatively light impurities were removed by means of the air separation unit. The industrial cameras (BFS-U3-51S5C-C, LUSTER LightTech Co., LTD., Beijing, China) with global shutters were designed and manufactured by FLIR and the supporting development tool was Spinnaker 2.6.0.160 (FLIR Systems, Wilsonville, OR, USA). The ring lights (RI15045-W) developed by OPT-Machine Vision were utilized to ensure the uniformity of imaging brightness. The resolution of RGB impurity-containing maize images were standardized to 600*600, which was beneficial to the feature learning network. In order to avoid the uncertain convergence direction and over-fitting conditions of the entire models caused by insufficient quantity of samples, data augmentation approaches were exerted to expand the datasets [65]. Specifically, we performed rotations, vertical mirror symmetry, horizontal mirror symmetry, adjustments of contrast and brightness, insertions of Gaussian noise and salt and pepper noise on the 1000 original images, as shown in Figure 7, and divided the impurity-containing maize datasets into a training set and test set, according to the ratio of 3:1 [66]. The adjustment and addition of brightness, contrast and noise enabled the model to have better robustness and greater adaptability to the image acquisition conditions.

The proposed model was regarded as the adaptive region proposal network (ARPN) and remaining detector network, which were trained through alternating optimization [67]. To be specific, the ImageNet-pre-trained models were used to initialize the feature learning network and the end-to-end training was performed on ARPN. Afterwards, the feature learning network was initialized again through the ImageNet-pre-trained models and the detector network was trained based on the proposals generated by ARPN. Eventually, both of the components shared the same convolutional layers and, sequentially, fine-tuned the layers unique to ARPN and the detector network, thereby forming a unified network [68]. The utilized deep learning framework was Pytorch 1.10, the version of Python was 3.7, the vision toolkit was Torchvision 0.11.1 and the strategy of stochastic gradient descent (SGD) was adopted to optimize the processes of parameter updating.

The comprehensive performance of the proposed tiny object detection network was measured through the evaluation indicators applied to the COCO datasets [69,70]. The AP in Figure 8 was the mean value of all mAPs (mean average precisions) when the IoU threshold was between 0.5 and 0.95 (with a value interval of 0.05), which indicated the localization capability of the model [71]. Among them, mAP was the average of the areas under the curves in the PR Graphs that corresponded to each object category. AP_50_ was the mAP (IoU threshold was 0.5) for all kinds of objects and AP_s_ could be defined as the AP for objects with sizes less than 64^2^ [72]. AR_100_ and AR_10_ (the range and value interval of IoU threshold were the same as those of AP) separately denoted the average of all mARs (mean average recall) for the *n* top-scoring detections after NMS (Non-Maximum Suppression) [73]. The mAR was twice the mean value of the areas under the curves in the Recall–IoU Graph corresponding to each object category. Similar to AP_s_, AR_s_ could be defined as the AR for objects with sizes less than 64^2^. Basic-ResNet101 and Basic-EfficientNetB7 represented replacing the backbone of the classic work Faster R-CNN with ResNet101 and EfficientNetB7, respectively. Basic + ARPN and Basic + Cross-stage integration mechanism individually signified the introduction of ARPN and Cross-stage integration on the basis of EfficientNetB7 as the feature learning network. The ablation experiments sequentially demonstrated the effectiveness of each improved component on the basic model, thereby revealing the superiority of the proposed model (ours), which exhibited stronger performance in various evaluation indicators (Figure 8A) [74,75]. The selection of EfficientNetB7 could significantly improve the tiny object detection capability, while the cross-stage integration mechanism and ARPN also had strong adaptability. Since the quantity of objects in each image was mostly no more than 10, the results of AR_100_ and AR_10_ were comparable. In addition, Figure 8B shows the average detection precision of various objects for different models when the IoU threshold was 0.5, reflecting the better equilibrium of the proposed model. Among them, the relatively lower average precision of the category Damaged might be caused by the similar appearance of partially damaged maize to normal kernels, and this also explained the higher average precision of category Weeds due to their more prominent profiles. Figure 9 exhibits the object detection outcomes on part of the images in the test datasets, including the predicted category and confidence score of the objects. The overall performance was consistent with the data pattern in Figure 8, which could reflect the object distribution in the maize cleaning processes.

## 4. Conclusions

In this study, we proposed a tiny object detection network specifically designed for harvested maize, to accurately identify and analyze the categories and distribution of impurities during the cleaning process. Firstly, on the basis of EfficientNetB7, a cross-stage integration mechanism was introduced to obtain feature matrices that contained spatial information and semantic descriptions. Then, the appropriate candidate boxes were generated through ARPN. Eventually, the classification and regression layers output the final detection results after adjusting the attribute information. The superiority of the proposed approach over the basic model was demonstrated through the ablation experiments on the constructed impurity-containing maize datasets and the effectiveness of each introduced component was illustrated as well. The introduction of the components individually or simultaneously enabled the model to have a stronger detection capability, which proved the compatibility between them. In addition, the proposed tiny object detection network also had better performance in actual continuous maize cleaning operations.

## 5. Future Direction

By virtue of the distribution information of various objects derived in the maize cleaning operations, the current study could provide significant references for the qualitative production. In the future, the structural design of the detection network will be optimized according to the comprehensive characteristics of more types of crops, so that it can be applied to more scenarios of cleaning operations.

## Figures and Tables

**Figure 1 foods-12-02885-f001:**
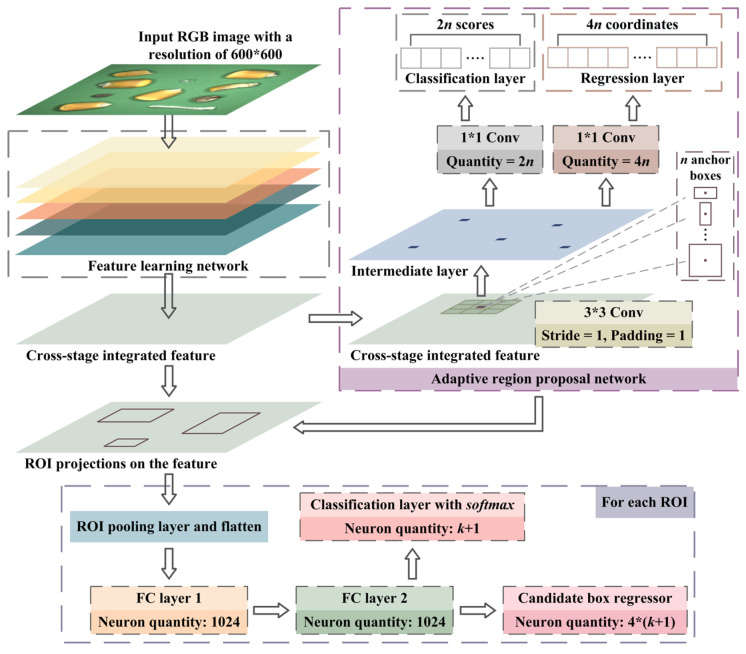
The tiny object detection network for impurity-containing maize images.

**Figure 2 foods-12-02885-f002:**
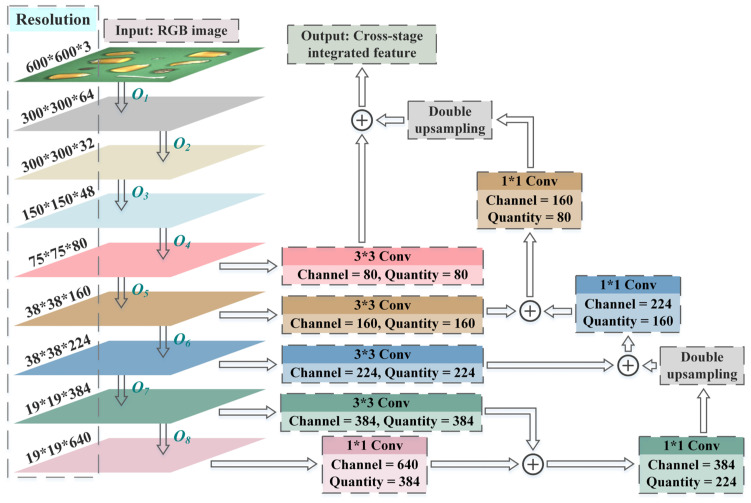
The cross-stage feature integration mechanism.

**Figure 3 foods-12-02885-f003:**
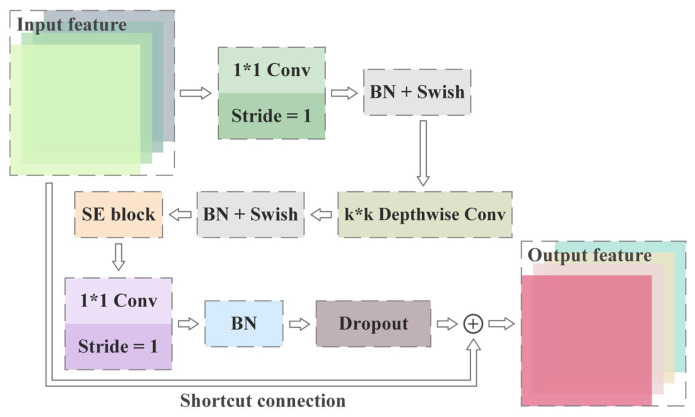
The detailed structure of MBConv operation.

**Figure 4 foods-12-02885-f004:**
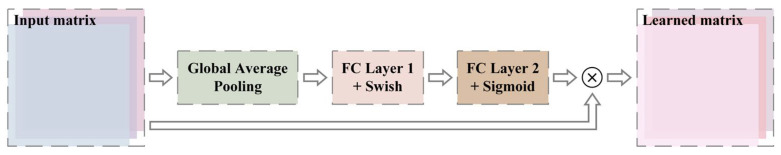
The module arrangement of the SE block.

**Figure 5 foods-12-02885-f005:**
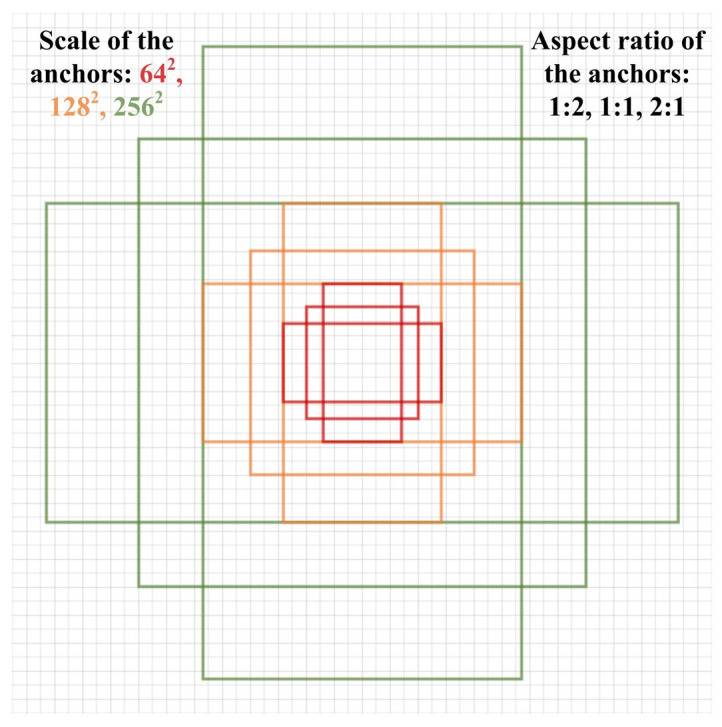
The area scale and aspect ratio of the anchor box.

**Figure 6 foods-12-02885-f006:**
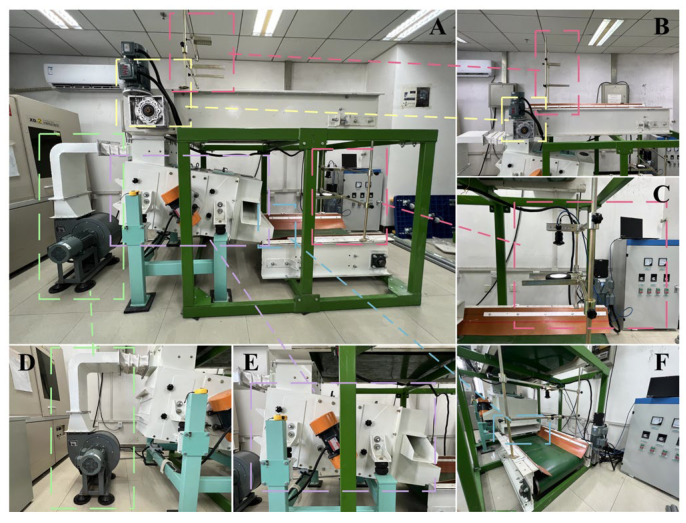
The multiphysics-coupled cleaning equipment (processed by Zhengzhou Wangu Machinery Co., Ltd., Zhengzhou, China) and its components. (**A**) The overall structure. (**B**,**C**) The image acquisition modules (pink frames) and the feed port (yellow frame). (**D**) The air separation unit (green frame). (**E**) The main part of the vibrating screen unit (purple frame). (**F**) The discharge port (cyan frame).

**Figure 7 foods-12-02885-f007:**
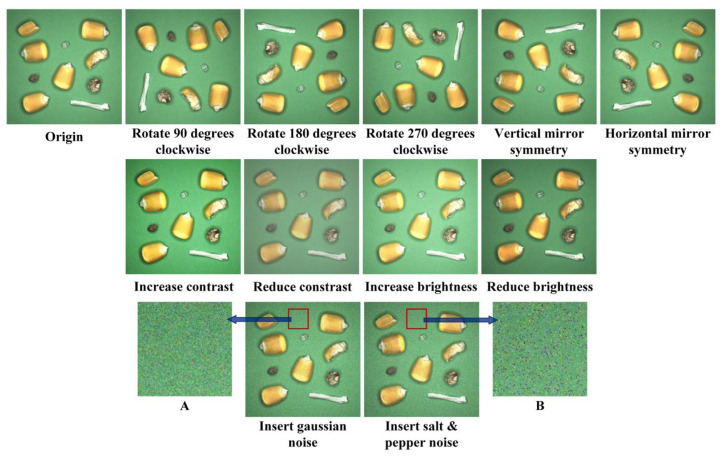
The impurity-containing maize images after data augmentation. (**A**) The inserted Gaussian noise. (**B**) The inserted salt and pepper noise.

**Figure 8 foods-12-02885-f008:**
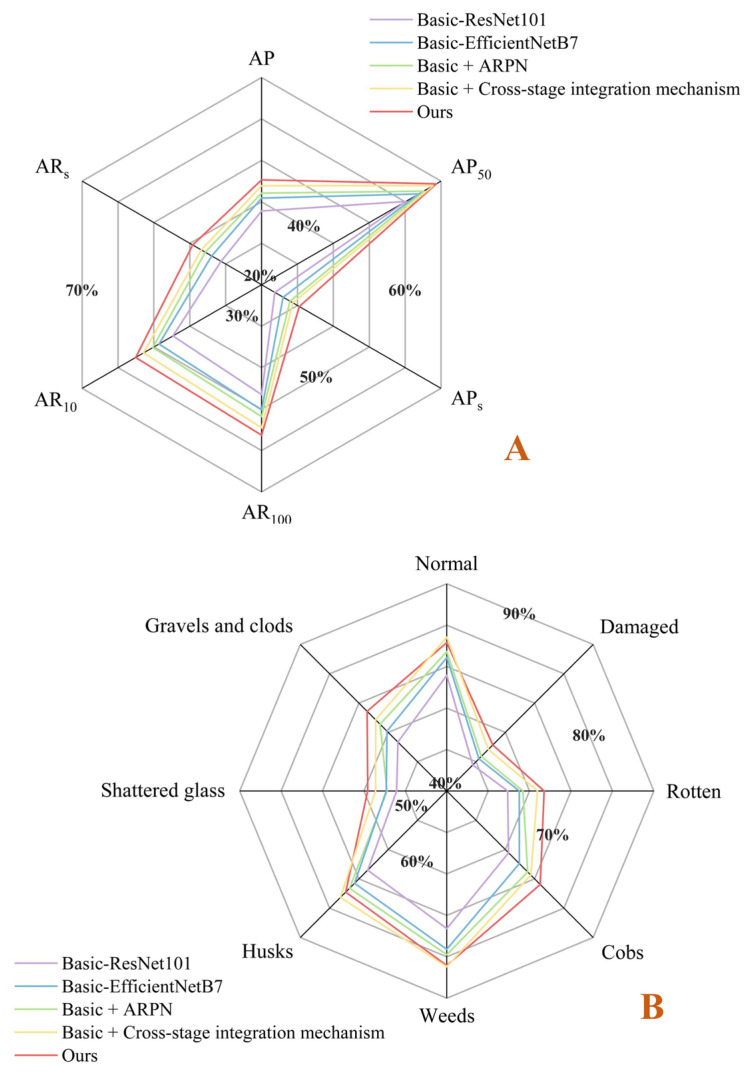
(**A**) Comparison of comprehensive detection performances among the proposed model and baselines. (**B**) The average detection precision of different models for various objects under the condition that the IoU threshold was 0.5 (corresponding to AP_50_).

**Figure 9 foods-12-02885-f009:**
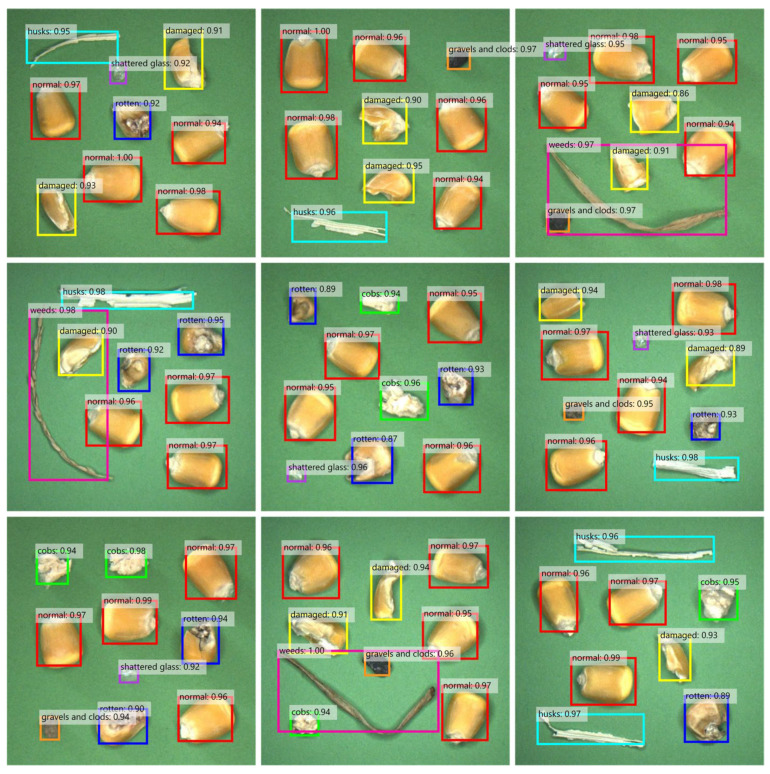
The object detection results on part of the images in the test datasets.

**Table 1 foods-12-02885-t001:** The specific components of the feature learning network.

Stage *i*	Operation *O_i_*	Resolution (Input) *H_i_***W_i_***C_i_*	Layers *L_i_*	Strides (First Layer)
1	3*3 Conv	600*600*3	1	2
2	MBConv1, k3*3	300*300*64	4	1
3	MBConv6, k3*3	300*300*32	7	2
4	MBConv6, k5*5	150*150*48	7	2
5	MBConv6, k3*3	75*75*80	10	2
6	MBConv6, k5*5	38*38*160	10	1
7	MBConv6, k5*5	38*38*224	13	2
8	MBConv6, k3*3	19*19*384	4	1

## Data Availability

The data presented in this study are available on request from the corresponding author.

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
