# Peer review of "A Tiny Object Detection Approach for Maize Cleaning Operations"

_foods, 2023, doi:10.3390/foods12152885_

Round 1
Reviewer 1 Report
1. Change the title of the paper
2. Add some more information about the proposed methodology in abstract
3. Add major contribution points at the end of the introduction section
4. Explain Figures 1, 2, 3, 4 and 5 in detail
5. Equation 1, 2,3, 4, 5 and 6 need to be explained in detail
6. Explain figure 6, 7, 8 and 9 properly
7. Add Future Direction section
8. Add some more explanation about the conclusion
9. Remove old references from the paper and add new updated references
Improve overall paper writing
Author Response
Reviewer #1:
We sincerely appreciate you for taking time to read our manuscript and giving us your valuable advice, especially the comments. They are constructive for improving our manuscript. We have addressed each of the comments as detailed below.
Comment 1: Change the title of the paper.
Response: Thanks for your valuable comments, which were critical to summarize the work of the entire paper, we change the title as follows:
Line 2:
A tiny object detection approach for maize cleaning operations.
Comment 2: Add some more information about the proposed methodology in abstract.
Response: Thanks for your constructive comments, the abstract was crucial for capturing the key information of the manuscript. Specifically, we supplemented the abstract as follows.
Lines 23-28:
The spatial information and semantic descriptions of feature matrices from different hierarchies could be fused through continuous convolution and upsampling operations. At the same time, taking into account the geometric properties of the objects to be detected and combining the images’ resolution, the adaptive region proposal network (ARPN) was designed and utilized to generate anchor boxes with appropriate sizes for the detectors, which would be beneficial to the capture and localization of tiny objects.
Comment 3: Add major contribution points at the end of the introduction section.
Response: Thanks for your valuable comments, which were greatly significant for the rigorousness of the manuscript. We made the following summaries to the major contributions of the study:
Lines 146-158:
The major contribution points could be summarized as follows:
- The EfficientNetB7 was introduced as the backbone of the feature learning network, and a tiny object detection network was proposed for analyzing the categories and distribution of impurities in the harvested maize based on the classic contribution Faster R-CNN;
- The designed cross-stage feature integration mechanism was able to fuse the semantic descriptions and spatial information of feature matrices from different hierarchies through continuous convolution and upsampling operations;
- Based on the geometric properties of the objects to be detected and the resolution of images, the adaptive region proposal network was able to generate appropriate anchor boxes for the detectors;
- The impurity-containing maize datasets was constructed to measure the comprehensive performance of the end-to-end tiny object detection network.
Comment 4: Explain Figures 1, 2, 3, 4 and 5 in detail.
Response: Thanks for your detailed comments, we added more details for these Figures as follows:
Lines 166-171:
The image feature learning network was used to extract the global features that contained multi-scale mappings; The adaptive region proposal network performed coordinate adjustment and classification of generated anchor boxes through continuous convolution; Eventually, the obtained high-quality candidate boxes were subjected to specific classification and location regression.
Lines 187-189:
Among them, the convolution operations with different receptive fields could simultaneously improve the expression ability of the model and adjust the dimension of the feature matrices.
Lines 213-215:
The utilization of depthwise convolution could effectively reduce the quantity of network parameters, which meant less memory consumption and faster computing speed.
Lines 231-233:
The global average pooling could downsample the matrices to the specified size, and the activation functions were able to improve the nonlinearity of the network.
Lines 255-257:
These were determined by conducting experiments on different categories of target contours in the impurity-containing maize images.
Comment 5: Equation 1, 2, 3, 4, 5 and 6 need to be explained in detail.
Response: Thanks for your detailed comments, which were significant for the expression accuracy of the manuscript. We checked and supplemented the descriptions of these Equations to ensure that each of the mentioned parameters was explained:
Lines 275-279:
The loss function of ARPN was shown in Equation (1), which was measured through dividing the sum of classification loss and regression loss by the quantity of mini-batch. Among them, was the capacity of each mini-batch. If the quantity of positive samples was less than 128, then the mini-batch would be supplemented with negative samples.
Lines 285-293:
The classification loss and regression loss were separately defined by the logarithmic and cumulative operations of Equation (2) and (3), and the revealed in Equation (4) was introduced as a robust loss function [26]. Furthermore, Equation (5) described the relationships among , and . and were the attribute information of the anchor and the coordinate-adjusted candidate box, respectively. was the attribute information of the ground-truth box corresponding to the current anchor [54]. The attribute information included the centre coordinates, width and height.
Comment 6: Explain figure 6, 7, 8 and 9 properly.
Response: Thanks for your detailed comments, we added more details for these Figures as follows:
Lines 344-346:
The multiphysics-coupled cleaning equipment removed impurities with a certain mass through two screens with different sizes and shapes, while relatively light impurities were removed by means of the air separation unit.
Lines 360-362:
The adjustment and addition of brightness, contrast and noise enabled the model with better robustness and greater adaptability to the image acquisition conditions.
Lines 402-404:
The selection of EfficientNetB7 could significantly improve the tiny object detection capability, while the cross-stage integration mechanism and ARPN also had well adaptability.
Lines 412-414:
The overall performance was consistent with the data pattern in Figure 8, which could reflect the object distribution in the maize cleaning processes.
Comment 7: Add Future Direction section.
Response: Thanks for your constructive comments, we added the Future Direction section as follows:
Lines 432-437:
By virtue of the distribution information of various objects derived in the maize cleaning operations, the current study could provide significant references for the qualitative production. In the future, the structural design of the detection network will be optimized according to the comprehensive characteristics of more types of crops, so that it can be applied to more scenarios of cleaning operations.
Comment 8: Add some more explanation about the conclusion.
Thanks for your meaningful comments, we organized and supplemented the content of conclusion as follows:
Lines 418-431:
In this study, we proposed a tiny object detection network specifically designed for harvested maize to accurately identify and analyze the categories and distribution of impurities during the cleaning processes. Firstly, on the basis of EfficientNetB7, a cross-stage integration mechanism was introduced to obtain feature matrices that contained spatial information and semantic descriptions. Then, the appropriate candidate boxes were generated through ARPN. Eventually, the classification and regression layers output the final detection results after adjusting the attribute information. The superiority of the proposed approach over the basic model was demonstrated through the ablation experiments on the constructed impurity-containing maize datasets, and the effectiveness of each introduced component was illustrated as well. The introduction of the components individually or simultaneously enabled the model to have a stronger detection capability, which proved the compatibility between them. In addition, the proposed tiny object detection network also had better performances in actual continuous maize cleaning operations.
Comment 9: Remove old references from the paper and add new updated references.
Thanks for your valuable comments, we removed relatively old references and supplemented newer works. Specifically, we supplemented the Introduction section as follows:
Lines 61-67:
As a kind of information carrier, images could provide research foundation and data resources for numerous fields [8]. Based on the hyperspectral, a rapid and cost-effective way was proposed to generate records of sediment properties and composition at the micrometer-scale [9]. Yuan et al. designed a compact proxy-based deep learning framework to perform highly accurate hyperspectral image classification with superb efficiency and robustness [10]. In addition, the maize kernel images supplied information supports to the classification tasks of planted cultivars [11].
Lines 77-80:
Relying on the powerful feature extraction capabilities, deep learning-based object detection technologies could adaptively capture the deep semantic information of images through the multi-structured network models, thus significantly improving the efficiency and accuracy of detection tasks [17, 18].
Reviewer 2 Report
This study aims to detect small objects specifically designed for images of harvested corn that contain impurities. The applicability of this research is in the identification and analysis of impurity categories in real time, during cyclic cleaning operations. However, there are a number of ambiguities that authors should clarify before considering a paper.
- the abbreviations used in the summary are not appropriate (e.g. for the type of neural networks)
- considering that the paper is not an overview, but scientific - the last sentence in the summary is also not appropriate
- further clarify the relationship between the variables listed in L163-165
- which represents A and B in Figure 7
Figure 8A, parameters APs about ARs are not clarified
in the conclusion, you state that you have validated the effectiveness of each introduced component - I suggest that you emphasize this validation with a special subtitle and express claims of "superiority" with concrete quantitative values.
I suggest correcting the last sentence of the conclusion. A scale-up in a real environment is proposed, in which the superiority of the proposed method compared to the standard one will be confirmed or refuted.
Sincerely
Author Response
Reviewer #2:
Comment 1: The abbreviations used in the summary are not appropriate (e.g. for the type of neural networks).
Response: Thanks for your detailed comments, which were critical to the rigorousness of the manuscript's presentation. We checked and revised the inappropriate abbreviation in the abstract as follows:
Lines 20-23:
On the basis of the classic contribution Faster Region Convolutional Neural Network, EfficientNetB7 was introduced as the backbone of the feature learning network, and a cross-stage feature integration mechanism was embedded to obtain the global features that contained multi-scale mappings.
Comment 2: Considering that the paper is not an overview, but scientific - the last sentence in the summary is also not appropriate.
Response: Thanks for your valuable comments, which were greatly significant for the rigorousness of the manuscript. We removed the inappropriate representation.
Comment 3: Further clarify the relationship between the variables listed in L163-165.
Response: Thanks for your valuable comments, which were crucial to the understanding of the study. There were no corresponding computational relationships among the listed variables, they were relatively independent, and the specific values were determined from the results of the experimental exploration in the classical work EfficientNetB7.
Comment 4: Which represents A and B in Figure 7. Figure 8A, parameters APs about ARs are not clarified.
Response: Thanks for your constructive comments. We supplemented the explanation of Figure 7. At the same time, it was checked and confirmed that the interpretations and explanations for each evaluation indicator. There was similar operational logic between APs and ARs, but there were no direct arithmetic relationships between them. Their specific calculation methods and meaning were referred to the specifications provided by the MS COCO datasets. The corresponding revisions were as follows:
Lines 364-365:
Figure 7. The impurity-containing maize images after data augmentation. (A) The inserted gaussian noise. (B) The inserted salt & pepper noise.
Lines 387-389:
AP50 was the mAP (IoU threshold was 0.5) for all kinds of objects and APs could be defined as the AP for objects with sizes less than 642 [67].
Lines 393-394:
Similar to APs, ARs could be defined as the AR for objects with sizes less than 642.
Comment 5: In the conclusion, you state that you have validated the effectiveness of each introduced component - I suggest that you emphasize this validation with a special subtitle and express claims of "superiority" with concrete quantitative values.
Response: Thanks for your valuable comments, we revised the conclusion as follows:
Lines 418-431:
In this study, we proposed a tiny object detection network specifically designed for harvested maize to accurately identify and analyze the categories and distribution of impurities during the cleaning processes. Firstly, on the basis of EfficientNetB7, a cross-stage integration mechanism was introduced to obtain feature matrices that contained spatial information and semantic descriptions. Then, the appropriate candidate boxes were generated through ARPN. Eventually, the classification and regression layers output the final detection results after adjusting the attribute information. The superiority of the proposed approach over the basic model was demonstrated through the ablation experiments on the constructed impurity-containing maize datasets, and the effectiveness of each introduced component was illustrated as well. The introduction of the components individually or simultaneously enabled the model to have a stronger detection capability, which proved the compatibility between them. In addition, the proposed tiny object detection network also had better performances in actual continuous maize cleaning operations.
Comment 6: I suggest correcting the last sentence of the conclusion. A scale-up in a real environment is proposed, in which the superiority of the proposed method compared to the standard one will be confirmed or refuted.
Response: Thanks for your detailed comments, we revised the conclusion as follows:
Lines 418-431:
In this study, we proposed a tiny object detection network specifically designed for harvested maize to accurately identify and analyze the categories and distribution of impurities during the cleaning processes. Firstly, on the basis of EfficientNetB7, a cross-stage integration mechanism was introduced to obtain feature matrices that contained spatial information and semantic descriptions. Then, the appropriate candidate boxes were generated through ARPN. Eventually, the classification and regression layers output the final detection results after adjusting the attribute information. The superiority of the proposed approach over the basic model was demonstrated through the ablation experiments on the constructed impurity-containing maize datasets, and the effectiveness of each introduced component was illustrated as well. The introduction of the components individually or simultaneously enabled the model to have a stronger detection capability, which proved the compatibility between them. In addition, the proposed tiny object detection network also had better performances in actual continuous maize cleaning operations.
Reviewer 3 Report
The title of manuscript is “A grain situation awareness approach driven by tiny object detection model for maize cleaning operations”.
I commented on the manuscript and the comments are presented below:
Part: Introduction.
The Introduction of the study provides with some general information about the techniques of the image analysis. On the other hand for several decades, studies on application of different type of techniques for image analysis have been conducted. I suggest supplementing the Chapter with additional information related to other new methods and devices in research of image analysis. Additional information contained in the Introduction chapter will make the aim of the study will clearly stated. „Hyperspectral and Multispectral Imaging in Dermatology”; „Hyperspectral imaging coupled with multivariate analysis and artificial intelligence to the classification of maize kernels”; “Proxy-based deep learning framework for spectral-spatial hyperspectral image classification: efficient and robust”. Additional information contained in the Introduction chapter will make the aim of the study will clearly stated.
Part: Material and methods
The Materials and Methods section provides the reader with not enough information to repeat the experiments conducted.
Part: Results
For the most part the Results section is well structured.
Part: Discussion
In the Discussion chapter, there is no full comparison and confrontation with the research of other authors in this area. The results were not fully discussed. A full discussion of the results obtained with other work in this field should be carried out in more aspects. I suggest supplementing the Chapter with additional information, for example: „Hyperspectral imaging coupled with multivariate analysis and artificial intelligence to the classification of maize kernels”, “Recent Advances in Multi- and Hyperspectral Image Analysis”; “Hyperspectral imaging: a novel, non-destructive method for investigating sub-annual sediment structures and composition”, ” Classification of black beans using visible and near infrared hyperspectral imaging”. In the discussion chapter contains information should be supplemented on discussing with the other items from the last years of publication including problems image analysis.
Part: Conclusion
The Conclusions chapter contains information obtained after conducting experiments but were no full comparison and confrontation with the research of other authors in this area.
Part: Reference.
The literature used is appropriate but should be supplementing about the items from the last years of publication about similar problem and methods of investigation.
Author Response
Reviewer #3:
We sincerely appreciate you for taking time to read our manuscript and giving us your detailed comments. They are very helping to improve our manuscript. We have revised the manuscript accordingly.
Comment 1: The Introduction of the study provides with some general information about the techniques of the image analysis. On the other hand for several decades, studies on application of different type of techniques for image analysis have been conducted. I suggest supplementing the Chapter with additional information related to other new methods and devices in research of image analysis. Additional information contained in the Introduction chapter will make the aim of the study will clearly stated. „Hyperspectral and Multispectral Imaging in Dermatology”; „Hyperspectral imaging coupled with multivariate analysis and artificial intelligence to the classification of maize kernels”; “Proxy-based deep learning framework for spectral-spatial hyperspectral image classification: efficient and robust”. Additional information contained in the Introduction chapter will make the aim of the study will clearly stated.
Response: Thanks for your valuable comments. On the basis of learning the innovative works in the comments, we supplemented the Introduction section with additional information related to other new methods and devices in research of image analysis.
Specifically, we perfected and revised the Introduction as follows:
Lines 61-67:
As a kind of information carrier, images could provide research foundation and data resources for numerous fields [8]. Based on the hyperspectral, a rapid and cost-effective way was proposed to generate records of sediment properties and composition at the micrometer-scale [9]. Yuan et al. designed a compact proxy-based deep learning framework to perform highly accurate hyperspectral image classification with superb efficiency and robustness [10]. In addition, the maize kernel images supplied information supports to the classification tasks of planted cultivars [11].
Lines 77-80:
Relying on the powerful feature extraction capabilities, deep learning-based object detection technologies could adaptively capture the deep semantic information of images through the multi-structured network models, thus significantly improving the efficiency and accuracy of detection tasks [17, 18].
Comment 2: The Materials and Methods section provides the reader with not enough information to repeat the experiments conducted.
Response: Thanks for your meaningful comments, it was extremely vital for the logical integrity of the manuscript. In the Materials and Methods section, the emphasis was placed on the description of the structure and principles of the proposed network. In section 3, we described the logic of network training and the setting of experimental parameters in detail as follows:
Lines 366-377:
The proposed model could be regarded as the adaptive region proposal network (ARPN) and remaining detector network, which were trained through alternating optimization [62]. To be specific, the ImageNet-pre-trained models were used to initialize the feature learning network, and the end-to-end training was performed on ARPN. Afterwards, the feature learning network was initialized again through the ImageNet-pre-trained models and the detector network was trained based on the proposals generated by ARPN. Eventually, both of the components shared the same convolutional layers and sequentially fine-tuned the layers unique to ARPN and detector network, thereby forming a unified network [63]. The utilized deep learning framework was Pytorch 1.10, the version of Python was 3.7, the vision toolkit was Torchvision 0.11.1, and the strategy of stochastic gradient descent (SGD) was adopted to optimize the processes of parameter updating.
Comment 3: In the Discussion chapter, there is no full comparison and confrontation with the research of other authors in this area. The results were not fully discussed. A full discussion of the results obtained with other work in this field should be carried out in more aspects. I suggest supplementing the Chapter with additional information, for example: „Hyperspectral imaging coupled with multivariate analysis and artificial intelligence to the classification of maize kernels”, “Recent Advances in Multi- and Hyperspectral Image Analysis”; “Hyperspectral imaging: a novel, non-destructive method for investigating sub-annual sediment structures and composition”, ” Classification of black beans using visible and near infrared hyperspectral imaging”. In the discussion chapter contains information should be supplemented on discussing with the other items from the last years of publication including problems image analysis.
Response: Thanks for your constructive comments. The advanced works mentioned in the comments had many similarities with this study in terms of technology. However, the proposed model was oriented towards the cleaning operation of harvested maize and was validated on a constructed impurity-containing maize datasets. Therefore, there were certain differences in the design objectives and application scopes compared to these works, thus preventing direct comparisons from being accomplished. We added these novel works to the Introduction section.
Specifically, we supplemented the Introduction section as follows:
Lines 61-67:
As a kind of information carrier, images could provide research foundation and data resources for numerous fields [8]. Based on the hyperspectral, a rapid and cost-effective way was proposed to generate records of sediment properties and composition at the micrometer-scale [9]. Yuan et al. designed a compact proxy-based deep learning framework to perform highly accurate hyperspectral image classification with superb efficiency and robustness [10]. In addition, the maize kernel images supplied information supports to the classification tasks of planted cultivars [11].
Lines 77-80:
Relying on the powerful feature extraction capabilities, deep learning-based object detection technologies could adaptively capture the deep semantic information of images through the multi-structured network models, thus significantly improving the efficiency and accuracy of detection tasks [17, 18].
Comment 4: The Conclusions chapter contains information obtained after conducting experiments but were no full comparison and confrontation with the research of other authors in this area.
Response: Thanks for your precious and detailed comments. The study was focused on specific scenarios, and although there were technical similarities with other existing works, there were certain differences in the practical application fields. In addition, the study was conducted on the constructed datasets, so it was not suitable for direct comparison with other existing works. We reorganized the Conclusion section to summarize and explain the contributions and results of the study.
Lines 418-431:
In this study, we proposed a tiny object detection network specifically designed for harvested maize to accurately identify and analyze the categories and distribution of impurities during the cleaning processes. Firstly, on the basis of EfficientNetB7, a cross-stage integration mechanism was introduced to obtain feature matrices that contained spatial information and semantic descriptions. Then, the appropriate candidate boxes were generated through ARPN. Eventually, the classification and regression layers output the final detection results after adjusting the attribute information. The superiority of the proposed approach over the basic model was demonstrated through the ablation experiments on the constructed impurity-containing maize datasets, and the effectiveness of each introduced component was illustrated as well. The introduction of the components individually or simultaneously enabled the model to have a stronger detection capability, which proved the compatibility between them. In addition, the proposed tiny object detection network also had better performances in actual continuous maize cleaning operations.
Comment 5: The literature used is appropriate but should be supplementing about the items from the last years of publication about similar problem and methods of investigation.
Response: Thanks for your valuable comments, we removed relatively old references and supplemented newer works.
Round 2
Reviewer 2 Report
All my comments have been taken into account and I recommend the work for the further publication process.